# A Low-Cost Open Source Device for Cell Microencapsulation

**DOI:** 10.3390/ma13225090

**Published:** 2020-11-11

**Authors:** Miriam Salles Pereira, Liana Monteiro da Fonseca Cardoso, Tatiane Barreto da Silva, Ayla Josma Teixeira, Saul Eliahú Mizrahi, Gabriel Schonwandt Mendes Ferreira, Fabio Moyses Lins Dantas, Vinicius Cotta-de-Almeida, Luiz Anastacio Alves

**Affiliations:** 1Laboratory of Cellular Communication, Oswaldo Cruz Institute, Oswaldo Cruz Foundation, 4365 Manguinhos, Rio de Janeiro 21045-900, Brazil; msmakeba@gmail.com (M.S.P.); lianamfc@gmail.com (L.M.d.F.C.); tatiane.barreto321@gmail.com (T.B.d.S.); aylajosma02@gmail.com (A.J.T.); 2Volta Redonda University Center—UniFOA, Av. Paulo Erlei Alves Abrantes, 1325-Três Poços, Volta Redonda 27240-560, Brazil; 3National Institute of Technology—INT, Rio de Janeiro Av. Venezuela, 82-Saúde, Rio de Janeiro 20081-312, Brazil; saul.mizrahi@int.gov.br (S.E.M.); gabriel.mendes@int.gov.br (G.S.M.F.); fabio.dantas@int.gov.br (F.M.L.D.); 4Laboratory on Thymus Research, Oswaldo Cruz Institute, Oswaldo Cruz Foundation, 4365 Manguinhos, Rio de Janeiro 21045-900, Brazil; vca@ioc.fiocruz.br; 5National Institute of Science and Technology on Neuroimmunomodulation (INCT-NIM), Oswaldo Cruz Institute, Oswaldo Cruz Foundation, 4365 Manguinhos, Rio de Janeiro 21045-900, Brazil

**Keywords:** alginate, 3D printer, microencapsulation, cell transplantation, cell therapy

## Abstract

Microencapsulation is a widely studied cell therapy and tissue bioengineering technique, since it is capable of creating an immune-privileged site, protecting encapsulated cells from the host immune system. Several polymers have been tested, but sodium alginate is in widespread use for cell encapsulation applications, due to its low toxicity and easy manipulation. Different cell encapsulation methods have been described in the literature using pressure differences or electrostatic changes with high cost commercial devices (about 30,000 US dollars). Herein, a low-cost device (about 100 US dollars) that can be created by commercial syringes or 3D printer devices has been developed. The capsules, whose diameter is around 500 µm and can decrease or increase according to the pressure applied to the system, is able to maintain cells viable and functional. The hydrogel porosity of the capsule indicates that the immune system is not capable of destroying host cells, demonstrating that new studies can be developed for cell therapy at low cost with microencapsulation production. This device may aid pre-clinical and clinical projects in low- and middle-income countries and is lined up with open source equipment devices.

## 1. Introduction

Currently, health technology and research equipment are crucial to improve medical care and pre-clinical experiments in low- and middle-income countries. In this context, several groups worldwide have developed technology using the concept of “open source equipment and medical devices”. This concept has been essential for the development of the software industry [1,2,3]. In keeping with these ideas, we have developed a low cost cell encapsulation device that can be easily constructed and can be used for cellular transplantation of different cell types.

Cell microencapsulation technology has been investigated and studied for over 36 years. This proposal was first described in 1964, when Chang microencapsulated mammalian erythrocytes in nylon based on the concept of developing artificial cells [4]. However, it was first used therapeutically in 1980, when Lim and Sun microencapsulated pancreatic beta cells in alginate in order to treat diabetic rats [5]. Recent progress in the field has resulted in biodegradable scaffolds that are able to retain and release cell contents in different anatomical locations. The microencapsulation process consists in the involvement of particles or biological materials, such as cells, at least one dimension less than 1.000 µm. Cells are microencapsulated when entrapped within a semipermeable polymer matrix (microsphere, microbead) at the micrometer scale [6,7]. The micrometer scale of microencapsulated cell implants is within the diffusion limits of many small molecules, such as glucose, amino acids, hormones, neurotransmitters and cytokines, while the pores of the encapsulating polymer are large enough to permit their influx. In addition, it is also possible to release small substances produced by microencapsulated cells as hormones and metabolites [7]. Another important point regarding microcapsule manufacture is the ability to create porosities smaller than the size that would allow a contact with components of the immune system, such as T cells, macrophages, antibodies and complement proteins, creating a barrier for interactions with surface antigens of microencapsulated cells [8]. A reduction in capsule size is considered to be one of the most important objectives in the microcapsules microenvironment, while allowing for a bidirectional nutrient, oxygen, and waste diffusion [9]. In addition, smaller capsules are more biocompatible than larger ones [10].

With the advent of cell therapy and tissue bioengineering, it became possible to treat lesions in various tissues present in pathologies and degenerative processes, which were previously untreatable. Thus, living cells have become an important tool in the advancement of therapeutic strategies with wide clinical applications [11]. However, the shortage of human organs that can be used as sources for obtaining target cells is an obstacle that has yet to be overcome for the success of some of these therapeutic approaches. In this context, microencapsulation is a widely studied technique for the accomplishment of cellular therapies, since the formed membrane allows for gas and vital nutrient permeability for cell maintenance, which does not activate immune system components, thus favoring its use in cellular transplantation [12]. Microencapsulation also protects cells from possible mechanical lesions, as well as from substances harmful to cellular viability, thus conferring greater survival conditions within the host [6,13].

In addition to application in cell therapies, particle microencapsulation has been studied and used in several areas for the transport of various bioactive compounds, such as drugs, vitamins, peptides, flavorings, dyes, essential oils, nutrients and pesticides [7,13]. Some basic aspects should be considered in the development of microencapsulated systems, such as the nature and stability of the material to be encapsulated, the microencapsulation process in itself, the characteristics of the encapsulating polymer and the product to be obtained.

The current relevance of microencapsulation derives from its use mainly as a therapeutic mechanism for treating a wide range of human diseases, such as diabetes, blood disorders, acute liver failure, spinal cord injury, and several types of cancer. Pancreatic islets, blood cells, hepatocytes, and stem cells are among the many cell types currently used for this strategy [14,15,16,17,18]. For this, different microencapsulation materials are being investigated. Microencapsulation materials comprise natural or synthetic polymers or blends, including collagen, gelatin, fibrin, polyphosphazenes, poly(acrylic acids), poly (methacrylic acids), copolymers of acrylic acid and methacrylic acid, poly(alkylene oxides), poly(vinyl acetate), polyvinylpyrrolidone, polyethylene glycol (PEG), polyethersulfone, polysaccharides such as agarose, cellulose sulfate, chondroitin sulfate, chitosan, hyaluronan, and copolymers, and blends of each [19,20]. The most widely researched microencapsulated cells are pancreatic beta cells within alginate/poly-l-lysine-based hydrogel microcapsules, which are currently applied in several studies for diabetes treatment [21,22,23]. The alginate membrane is a biodegradable polymer, derived from brown algae, and has been used in xenotransplantation procedures to generate protection for the transplanted cells against attack by the immune system, allowing for the diffusion of nutrients, oxygen and metabolic products, in order to maintain cellular physiology. The improvement of this technique may favor the development of therapies without requiring immunosuppressive drugs [24,25,26]. Our group develops a research area focused on the treatment of liver diseases through hepatocyte transplantation. In view of the difficulties of obtaining human cells to perform the therapeutic procedure, as well as to avoid an immune response against the injected cells, we have developed a low cost device for cellular microencapsulation produced in a 3D printer or associated with syringes. Thus, this study describes the encapsulation device mechanism and the formed “membrane” characteristics concerning microcapsule manufacture using sodium alginate.

## 2. Materials and Methods

### 2.1. Cell Line

Human hepatocellular carcinoma cells (HepG2) were maintained following standard mammalian cell culture practices (ATCC, Manassas, VA, USA). The cells were cultured in Roswell Park Memorial Institute medium (RPMI-1640) (Sigma-Aldrich, St. Louis, MO, USA) and supplemented with 10% fetal bovine serum (FBS) (Gibco Laboratories, Grand Island, NY, USA). They were then incubated at 37 °C in a 5% CO_2_ humidified environment in cell culture dishes until 70–80% confluency. For the experiments, the cells were trypsinized with a trypsin solution at 0.025% containing 0.4% EDTA (ethylene diamine tetraacetic acid) (Sigma-Aldrich, St. Louis, MO, USA).

### 2.2. Device

Two types of devices were created for cellular encapsulation. The first used a long-stay peripheral vein puncture equipment of two different sizes, the best-sized (24G) metal cutting part attached to the polypropylene part of the more calibrated device (16G). A 40 mm × 12 mm needle is located on the side, which allows N_2_ to enter, forming a parallel air system, which reduces the diameter of the micro-paste to be formed in the alginate solution. This system is connected to a 1 mL syringe containing cells diluted in a sodium alginate solution (Figure 1A). The second equipment parts were designed and printed using a 3D printer (Appendix A), in order to improve syringe and needle attachments to the previous system (Figure 1C). The 3D model file is available at an online link: https://drive.google.com/file/d/181ZF6c_oanhCNiYoaxchdh3d78l3yQdC/view?usp=sharing.

### 2.3. Cell Encapsulation

To test and optimize the use of the encapsulation device, the hepatocarcinoma HepG2 cell line was used as the biological material to be encapsulated. Cells were cultured at 2 × 10^6^ cells/mL in RPMI medium supplemented with 10% FBS (Gibco Laboratories, Grand Island, NY, USA) and incubated at 37 °C under a 5% CO_2_ atmosphere. After culturing, the cell-formed pellet was resuspended in a HEPES-EDTA buffer pH 7.0 and sodium alginate solution, diluted in 0.9% NaCl pH 7.0 at a final 1 to 4% alginate concentration (ideal concentration, 3%) to obtain a minimum cell density of 1 × 10^6^ cells/mL. Then, the solution was aspirated with a 1-mL syringe and placed in the encapsulation prototype, coupled to the parallel N_2_ system at 10 L/min to obtain more uniform microcapsules with less than 500 µm. Microdrops are then formed and fall into a Petri dish containing 0.1, 0.5 and 1.0 M calcium chloride solution at pH 7.0 for microcapsule polymerization. It should be noted that the plates were kept in constant movement, avoiding microcapsule overlap. Subsequently, the microcapsules were washed in RPMI-1640 medium containing 10% FBS and centrifuged for 3 min at 1000 rpm at 4 °C. The wash solution was then removed and the capsules were maintained in the culture medium at 37 °C under a 5% CO_2_ atmosphere.

### 2.4. Sodium Alginate “Membrane” Porosity

The morphology and diameter of the sodium alginate microcapsules were evaluated by optical microscopy (Nikon Eclipse TE-2000S, Tokyo, Japan) coupled to 10× and 20× lenses using 2 µM 20, 70 and 150 KDa FITC-dextran (fluorescein isothiocyanate) (Sigma-Aldrich, St. Louis, MO, USA). Briefly, the microcapsules were incubated with the different molecular weight dyes and evaluated at different time intervals (zero, 24 and 48 h) in an *in vitro* culture. Fluorescence was analyzed by confocal microscopy (Leica DMi8, Wetzlar, Germany).

### 2.5. Viability Assessment of Encapsulated Human Hepatocellular Carcinoma Cells (HepG2)

Fluorescence cell markers were used to determine cell viability. Calcein-AM is a cell-permeant dye that can be used to determine cell viability in most eukaryotic cells. In live cells, non-fluorescent calcein-AM is converted to a green-fluorescent calcein after acetoxymethyl ester hydrolysis by intracellular esterases and confined in the intracellular environment. For this assay, HepG2 cells were incubated with 2 µM calcein-AM (Molecular Probes, Eugene, OR, USA) at 37 °C for 30 min. The cells were then washed in two steps with RPMI medium at 50 g at 4 °C for 5 min and subsequent analyzed by fluorescence microscopy. As the most important factor for evaluating cell viability is to verify plasma membrane integrity, propidium iodide (PI), a fluorescent compound impermeable to the plasma membrane, was used for this purpose. This compound is an intercalating 668.39 Da molecule of paired nucleic acid bases that reaches the intracellular environment after membrane damage. The hepatocytes were incubated with 1 µg/mL PI (Sigma-Aldrich, St. Louis, MO, USA) for 5 min and washed in two steps with culture medium at 50 g 4 °C for 5 min, followed by three washes with the RPMI medium at 50 g medium at 4 °C for 5 min and analyzed by fluorescence microscopy. A similar protocol was performed with YO-PRO1 (Molecular Devices, San José, CA, USA), a 629 kDa fluorescent dye, and trypan blue (873 KDa) (Sigma-Aldrich, Korea) for cell viability assessments.

### 2.6. Statistical Analysis

All numerical results are presented as an arithmetic mean ± standard deviation (SD). All experiments were performed on at least three different days. The D’Agostino and Pearson normality test was used to assess data normality distribution. If the data followed a Gaussian distribution, an appropriate parametric test was applied; otherwise, an appropriate non-parametric test was applied. The applied tests are specified in the figure legends. *p* values of 0.05 or less were considered significant. Graphs and statistical analyses were performed using the GraphPad Prism version 5 software (GraphPad Software, San Diego, CA, USA).

## 3. Results

### 3.1. Cell Encapsulation Prototype Development

The prototype was developed using long-stay peripheral venipuncture devices of two different sizes, the smaller metal gauge cutter (size 24G) connected to the syringe containing the sodium alginate solution (Figure 1A(X)). Externally, the polypropylene structure of the larger diameter device (size 16G) was used, securing them and maintaining a longer tip of the metal needle (Figure 1A(Y)). A lateral opening was created for needle coupling, allowing nitrogen to enter, generating a tension on the surface of the formed microcapsule, thus reducing its diameter (Figure 1A(Z)). The microtiter drops into a Petri dish containing a calcium chloride solution, inducing sodium alginate polymerization. Then, the device was perfected, using parts printed by employing a 3D printer. The syringe body of this new device is secured in a longitudinal holder and its plunger is pressed by a threaded part which allows the alginate solution to fall through the turn, as depicted in Figure 1C. The syringe is connected to the metal needle of the previous 24G device, and the larger gauge external 16G device attached to a printed part (Appendix A), which allows the influx of nitrogen, thereby engaging the part to the gas bulb hose.

Some variables were determined for the standardization of the cellular encapsulation, as follows: needle and syringe size to be chosen, nitrogen gas flowmeter pressure, distance between the needle and 16G calcium chloride solution and sodium alginate and calcium chloride concentrations. Two different gauge needles were tested. The first measured 25 mm × 0.7 mm, whose produced capsules displayed a diameter of more than 2 mm. The second needle measured 14 mm × 0.38 mm (used for intradermal and subcutaneous administrations in humans) and displayed a diameter ranging from 700 to 1300 µm. The size of syringes, 1, 3 and 5 mL, was also assessed. The capsules in the 3 and 5 mL syringes presented many deformities, while the 1 mL syringe displayed higher homogeneity. Thus, a device that allowed for the insertion of a parallel gas system with nitrogen, used in food production and freezing, as well as in cell conservation, was created. The gas flow was tested at 1, 3 and 5 L/min in a flowmeter, coupled to a gas cylinder, in the first prototype. The 5 L/min flow reduced the diameter of and created the most homogeneous microdrops, forming capsules ranging from 500 to 650 µm. However, when testing the pressure in the second prototype, the capsules were not different from the previous prototype, requiring that the gas flow be increased to 10 L/min. This flow allowed for formation of capsules smaller than 500 µm without any shape change. The sodium alginate microcapsules were evaluated by scanning electron microscopy-SEM (Hitachi TM 3000) fixed on carbon tapes at 5 kV, with magnification from 120× (A), 150× (B) and 250× (C). (Appendix A)

Assessment of distance between the needle and the calcium chloride solution indicate that the 10 cm distance led to larger diameter capsules (710 ± 135 µm), as compared to 20 (364 ± 65 µm) and 30 cm (593 ± 136 µm) distances (Figure 2). Moreover, capsules were more homogeneous at 20 cm distance in height (Figure 2B). However, at 30 cm, loss of material around the Petri dish and varied-sized capsules were noted. Thus, the ideal distance for this type of prototype was set at 20 cm.

Capsules at a 1% to 4% sodium alginate concentration were tested. Membrane variability was observed under a light-field and grayscale microscopic analyses (Figure 3). At 1% and 2% concentrations, a thinner membrane (Figure 3A,B) was observed when compared to concentrations of 3% and 4% (Figure 3C,D). The capsules produced with 4% alginate had a thicker-looking membrane when viewed under light microscopy (Figure 3C,D). Therefore, 3% concentration was chosen for *in vitro* evaluations (Figure 3C). All sodium alginate concentrations were tested using the second prototype, allowing for the formation of capsules under 500 µm. Concerning these capsules, a significant difference between 1% and 4% concentrations was observed (Figure 3D).

As contact with calcium chloride is required for alginate polymerization and capsule formation, we assessed the influence of different concentrations of this solution on membrane polymerization. The capsules produced in 0.1 M calcium chloride presented size variability (Figure 4A), with diverse forms (Figure 4B), presence of membrane artifacts (Figure 4C) and with a larger diameter (718.5 ± 26.87 µm). In contrast, 0.5 M polymerized capsules (Figure 4E) presented smoother and homogeneous forms (Figure 4D), decreased diameter (616.9 ± 16.21 µm) and remained unchanged for up to seven days in the culture medium (Figure 4E). These experimental tests were performed using the first prototype. Concerning the second prototype, diameters were greatly reduced at 0.5 M (Figure 3E), and no significant difference between the 0.5 and 1 M concentrations was noted.

After variable evaluations and technique standardization, we performed a blind experiment with three different operators, in order to verify encapsulation technique reproducibility. Each operator performed all procedures three times on different days and the diameter count of 20 microcapsules per day was acquired (Figure 5). Although the results of the third operator were statistically different from the other operators, the diameters of the microcapsules were very similar, ranging from 400 to 650 µm. This slight difference may have occurred due to the lack of experience of the third operator with handling and producing microcapsules using the tested device. This factor is also important, as we have demonstrated that our device can be handled by any individual with no previous experience, which can be improved with handling time. Thus, the low-cost device is deemed simple and easy to use.

To evaluate the cellular viability of HepG2 cells, labeling with a 2 µM YO-PRO solution was carried out. Encapsulated living cells were unmarked, in contrast to the control with capsules containing dead cells (Figure 6).

In addition, we evaluated cell viability using calcein-AM and propidium iodide (IP). For this, the live HepG2 and HepG2 cells killed with detergent (5% Tween) were encapsulated at a density of 1 × 10^6^, grown with RPMI medium (Sigma Aldrish) supplemented with 10% fetal bovine serum (Sigma Aldrish) and kept in culture at 37 °C, 5% CO_2_ for 24 h. Then, we incubate the cells with calcein-AM (2 µM) for 30 min, calcein-AM when hydrolyzed by intracellular esterases generates a molecule that binds to intracellular calcium resulting in a high green fluorescence, thus defining living cells present in the medium. Propidium iodide, by binding to DNA, can easily identify non-viable cells, since it is not transported by the plasma membrane; this dye can only mark DNA from broken or damaged cells in the plasma membrane. Thus, we incubate our encapsulated cells with Propidium Iodide (1 µg/mL) for 5 min. For fluorescence visualization, we used the fluorescence microscope at 495/515 nm wavelength for calcein-AM and 535/617 nm for propidium iodide. In this experiment, we were able to visualize cells that were encapsulated alive by fluorescing strongly with calcein-AM and low fluorescence for propidium iodide, compared to dead encapsulated cells we can easily identify that they were not marked with calcein, but with propidium iodide after 24 h of cultivation (Figure 7). The Viability of microencased cells (HepG2) up to 72 h is illustrated in Appendix A.

### 3.2. Microcapsule Characteristics

Microcapsules were labeled with FITC to assess membrane shape and porosity. Permeability to 20-KDa FITC-dextran was observed in capsules containing HepG2 hepatocytes (Figure 8D). For the 70-KDa FITC-dextran, decreased permeability, allowing internal cell visualization, was noted (Figure 8E). Regarding the 150-KDa FITC-dextran, no capsule permeability was observed under fluorescence microscopy (Figure 8F). This demonstrated that the alginate membrane displays porosity for components up to 70 KDa, which would allow for HepG2 cellular therapy.

## 4. Discussion

Alginate is composed of natural polymers extracted from brown algae, displaying biocompatible, biodegradable, nontoxicity and easy availability characteristics [24,27]. This polysaccharide is composed of 1,4′-β-d-mannuronic acid (M) and α-l-guluronic acid (G) units, and simple gelation occurs when divalent cations, such as Ca^2+^, Sr^2+^, or Ba^2+^ interact with G monomers, forming ionic bridges between adjacent alginate chains [28]. Researchers worldwide have explored possible alginate applications such as coating material and applied to the preparation of controlled-release drug-delivery systems, such as microspheres, beads, pellets, gels, fibers, membranes, among others [6,24]. Alginate-based microcapsules can be coated with a permselective layer that allows for the diffusion of small molecules and proteins while also providing immune privilege by blocking antibodies and cells that contribute to immune rejection [29]. However, a persistent issue in application of alginate is the different degree of biocompatibility, which seems to be laboratory-dependent [25,30]. Different factors such as the use of different types of alginates [31], the type of coating [13] and variations in the purity of alginate have been shown to be a major cause of the variations in success of the capsules in terms of biocompatibility and acceptance by the host [25,31]. Also, items such as capsule porosity is a criteria for cell survival [32] as well as stiffness that might influence cell differentiation [33]. Purification of alginate is reported to reduce inflammatory responses against alginate based capsules but many groups have difficulties in reliably producing ultrapure alginates [34,35]. Another issue is that many used procedures to purify alginate have been published [36], but techniques to predict whether the purification is efficacious are lacking. In this work, the composition of the alginate that we used was approximately 61% mannuronic acid and 39% guluronic acid. The molecular weight of this product is approximately 240 kDa, according to the manufacturer’s information.

Cell microencapsulation holds promise for the treatment of many diseases by the continuous delivery of therapeutic products [13,37]. Clinical trials using islets encapsulated in alginate microcapsules have shown some promise as a treatment for type I diabetes [38,39].

Nevertheless, there is currently a significant shortfall between the number of patients who need lifesaving transplants and the number of donated human organs. In this context, xenotransplantation addresses this relevant matter and the application of xenogeneic cells has become promising to treat several disorders, including neurodegeneration and liver failure [37,38]. While immunologic incompatibilities have presented a persistent impediment to their use, encapsulation may represent a way forward for the use of cell-based xenogeneic therapeutics without the need for immunosuppression [39,40,41].

Our device was efficient in the production of sodium-alginate capsules by the parallel air system with diameters compatible to those described in the literature, between 400 and 500 µM. Moreover, the porosity of the produced capsules is compatible with the passage of components up to 70 kDa in size, allowing for proper secretion of metabolites, such as albumin and urea, in the case of employing hepatocytes. Some reports use a syringe system and different concentrations of the sodium alginate solution for microcapsule crosslinking. The diameter of the capsules produced in these studies can range from 10 to 1000 µm [42,43,44,45]. However, a complete device is described herein, so that groups wishing to work with cellular microencapsulation can produce their own low-cost equipment, with reproducible results. We provide the description and measurement of each piece that makes up the device for 3D printing. In addition, we demonstrate the reproducibility of capsules with specific diameters after device manipulation by three different operators. Both microcapsule size and morphology were very similar among the operators, reinforcing the fact that our device can be manipulated by different people to obtain very close and satisfactory results.

Chantel Farias and collaborators used a microencapsulation system with parameters similar to ours to microencapsulate HepG2 cell and human glioblastoma cell (U-87) in 2018. The authors observed significant loss of viability post-24 h incubation of the microencapsulated HepG2 cells, probably due to the spheroid formation detected after a day, limiting diffusion transport of oxygen and nutrients due to a stagnant microenvironment [43]. In our system, we have shown that cells remain viable 24 h after encapsulation, and further studies are warranted to define the viability of cells freed from the microcapsules, by approaching their ability to proliferate and differentiate.

Bressel and collaborators also developed a microencapsulation device using a parallel air system and adaptations similar to those described in this work. The group reports the optimization of a cost-benefit protocol obtaining promising results *in vitro* from cellular microencapsulation. Such results include maintaining viability in culture for 4 weeks without any signs of necrosis, and protein diffusion was observed during this period. In addition, encapsulated cells under the conditions described were able to secrete an active enzyme even after four weeks, thus becoming potentially compatible with therapeutic protein delivery [46].

Altogether, our experiments provide the basis for applying a simple and low-cost technique for cell microencapsulation, by employing a device suitable for production in 3D printers. Moreover, it further supports the use sodium alginate-based microcapsules amenable to be further assessed in cell therapy pre-clinical studies.

## Figures and Tables

**Figure 1 materials-13-05090-f001:**
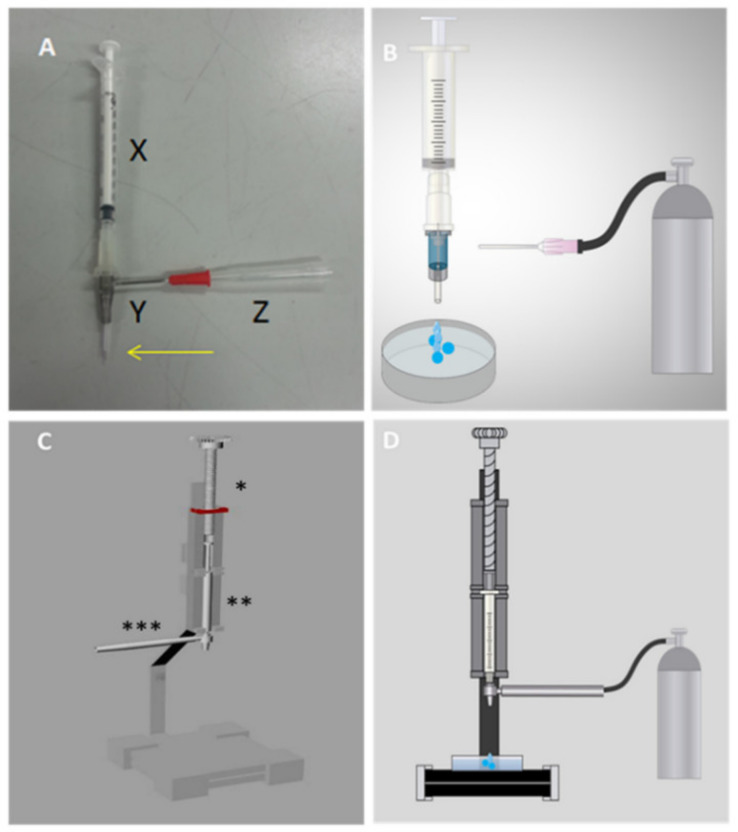
Prototype developed to carry out cellular encapsulation through a parallel air system. (**A**) First prototype produced with a long-stay peripheral venous puncture device (X = 1 mL syringe containing the sample in sodium alginate; Y = long-stay peripheral venous devices 16G on the outside and 24G on the inside; = needle connected to the nitrogen gas hose). The arrow highlights the end of the metallic needle 24G exposed to the environment; (**B**) first cellular encapsulation prototype coupled to the nitrogen gas bullet; (**C**) second complete prototype, prepared using a in 3D printer; * part that positions the syringe with the sample to be encapsulated; ** 1 mL syringe with sodium alginate solution; *** piece that connects to the parallel air system; (**D**) second cellular encapsulation prototype coupled to nitrogen gas.

**Figure 2 materials-13-05090-f002:**
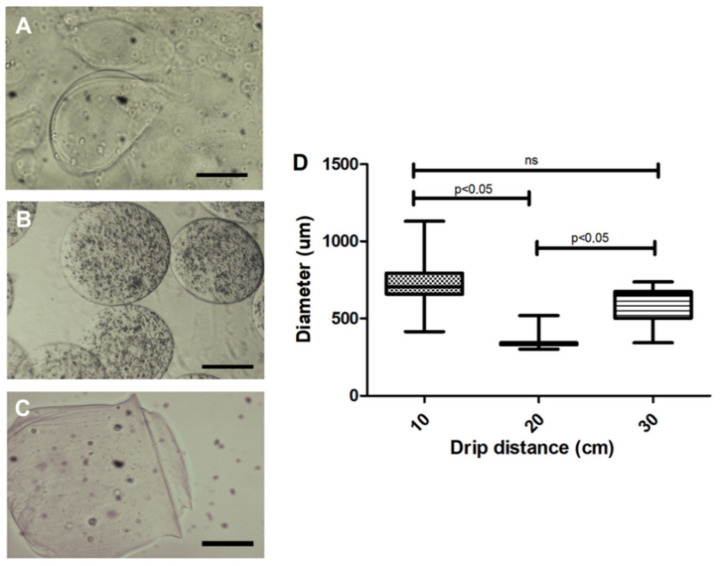
Effect of the microdrop distance of the 3% sodium alginate solution on microcapsule morphology. (**A**) A 10 cm distance presented variability in shape and larger diameters (710 ± 135 µm); (**B**) A 20 cm distance presented a more homogeneous shape and smaller diameter (364 ± 65 µm); (**C**) A 30 cm distance cm displayed an average diameter of 593 µm with more heterogeneous forms; (**D**) Quantification of the diameter of the capsules in relation to the distance of the drip. Photos recorded using light field optical microscope; Five independent experiments were performed in triplicate. 200 µm bar. Values were considered statistically significant when *p* < 0.05. ns = non-significant.

**Figure 3 materials-13-05090-f003:**
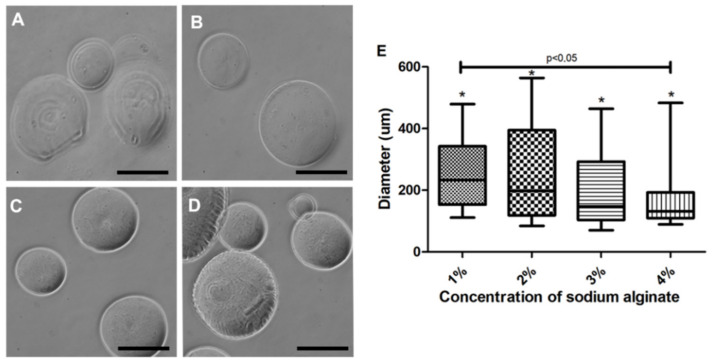
Morphological aspect of the microcapsules produced with the second prototype at a 20 cm distance. (**A**) 1% sodium alginate led to an altered membrane when viewed under light microscopy; (**B**) 2% sodium alginate presented a diameter variability (258 ± 154 µm); (**C**) 3% sodium alginate produced capsules with the smoothest and most homogeneous membrane; (**D**) 4% sodium alginate produced smaller capsules (166 ± 97 µm) with a more heterogeneous membrane. (**E**) Characterization of the diameter of the capsules produced with different sodium alginate concentrations. Photos recorded using light-field microscope in gray scale (Nikon); five independent experiments were performed in triplicate. * Values were considered statistically significant when *p* < 0.05. ns = non-significant. 200 µm bar.

**Figure 4 materials-13-05090-f004:**
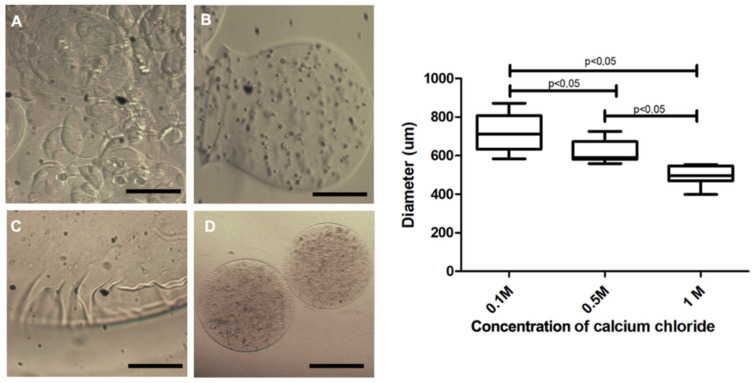
Capsule morphology produced with 3% sodium alginate and different calcium chloride concentrations. In (**A**,**B**), capsules polymerized in 0.1 M calcium chloride exhibited format variability; (**C**) displays a representative image of the presence of artifacts in the membrane of the capsules polymerized in calcium chloride 0.1 M; (**D**) capsules polymerized in 0.5 M calcium chloride, presenting a smooth membrane and diameter smaller than 0.1 M. Photos recorded using a light field microscope (Nikon); evaluation of the size of the 3% sodium alginate capsules in the polymerization process in 0.1 M, 0.5 M and 1.0 M calcium chloride. Data are plotted as mean and standard deviation. Five independent experiments were performed in triplicate using the first prototype. Values were considered statistically significant when *p* < 0.05. 200 µm bar.

**Figure 5 materials-13-05090-f005:**
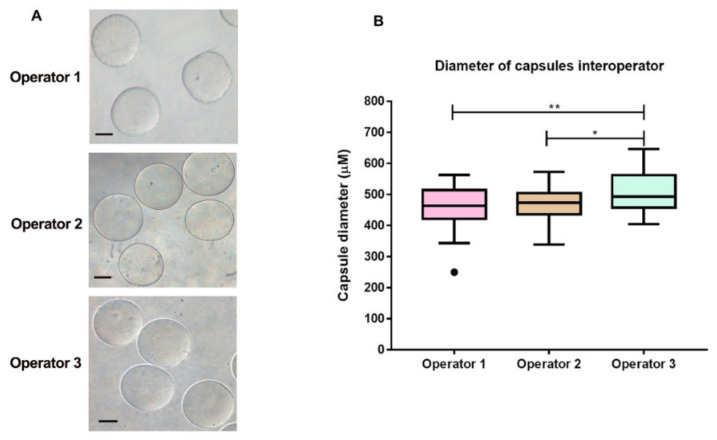
Reproducibility of the microcapsule production. (**A**) Morphological aspect of the capsules produced by the three operators in three independent experiments on different days. Scale bar presents 200 µm. (**B**) Comparative evaluation of the diameter of the capsules produced by three blind operators using the cellular encapsulation equipment printed using a 3D printer. The box plots illustrate the production of microcapsules using the low cost device by the three different operators. Reproducibility of sodium alginate microcapsule production was verified microcapsules by three independent experiments with three different operators. Each operator produced a total of 20 capsules and the diameter of each was measured. Box plot explanation: upper horizontal line of box, 75th percentile; lower horizontal line of box, 25th percentile; horizontal bar within box, median; upper horizontal bar outside box, 90th percentile; lower horizontal bar outside box, 10th percentile. Circles represent outliers. The medians significantly ** = *p* < 0.005 and * = *p* < 0.05.

**Figure 6 materials-13-05090-f006:**
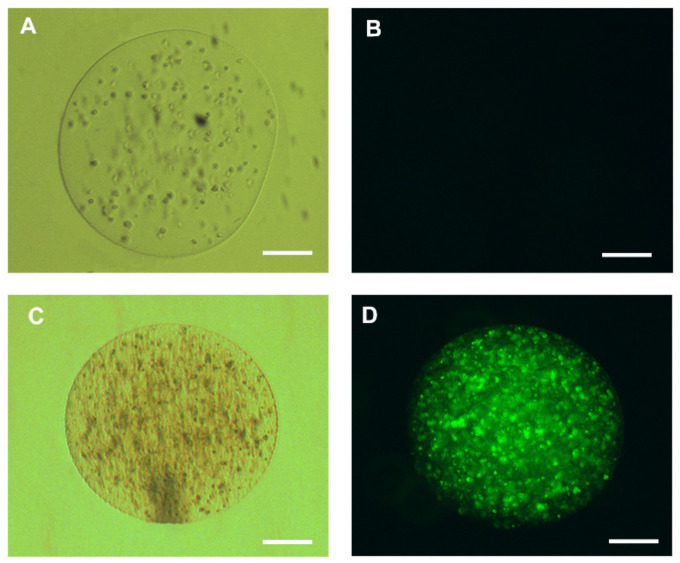
Representative cell viability test of human hepatocellular carcinoma cells (HepG2) cells encapsulated in 3% sodium alginate; (**A**) live encapsulated hepatocytes maintained for 24 h in Roswell Park Memorial Institute (RPMI) medium with 10% fetal bovine serum (FBS); (**B**) HepG2 cells encapsulated in 2 µM YO-PRO1 solution. The living cells do not display any dye labeling, represented by the absence of fluorescence inside the capsule; (**C**) dead encapsulated HepG2 cells maintained for 24 h in RPMI medium with 10% FBS visualized on the clear field and (**D**) HepG2 cells displaying fluorescence after labeling with 2 µM of YO-PRO1; 200 µm bar.

**Figure 7 materials-13-05090-f007:**
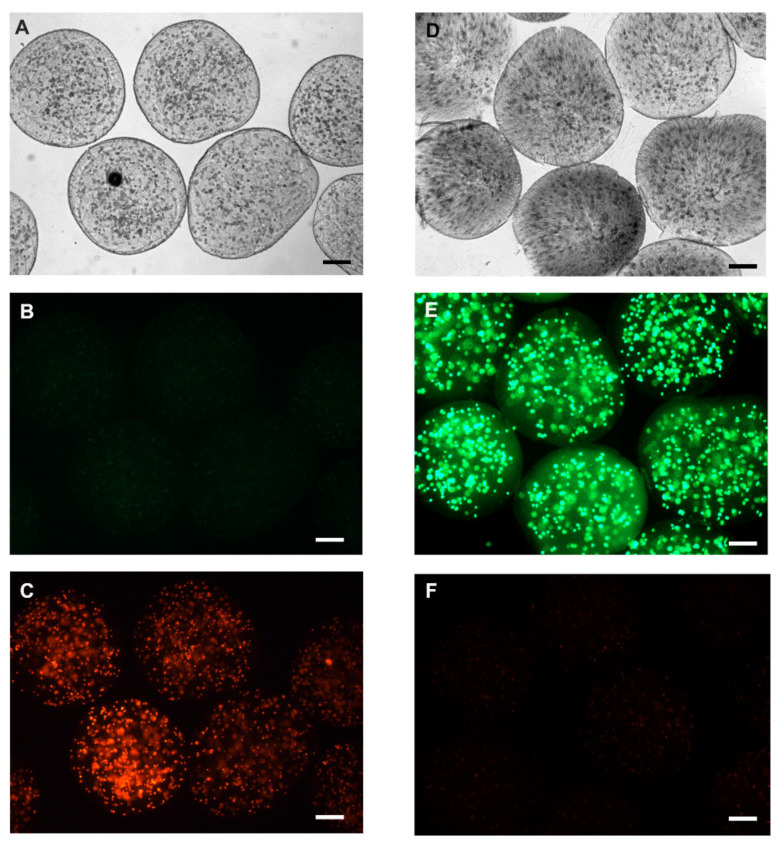
Representative fluorescence microscopy image with calcein-AM and propidium iodide. Initially, an assay for the negative cell viability control was performed. The cells were microencapsulated after incubation with 0.01% Triton X detergent for 5 min before encapsulation in the alginate membrane. (**A**) Clear field microscopy photo of dead encapsulated cells; (**B**) negative labeling for calcein-AM, and (**C**) cells with positive labeling for propidium iodide. Cell viability after encapsulation was verified by encapsulating HepG2 cells with viability greater than 90%. (**D**) clear field representation of cellular microencapsulation; (**E**) positive labeling for calcein-AM, representing cell viability after encapsulation, and (**F**) some cells labeled with propidium iodide. Representative photo of three experiments. Scale bar presents 100 µm.

**Figure 8 materials-13-05090-f008:**
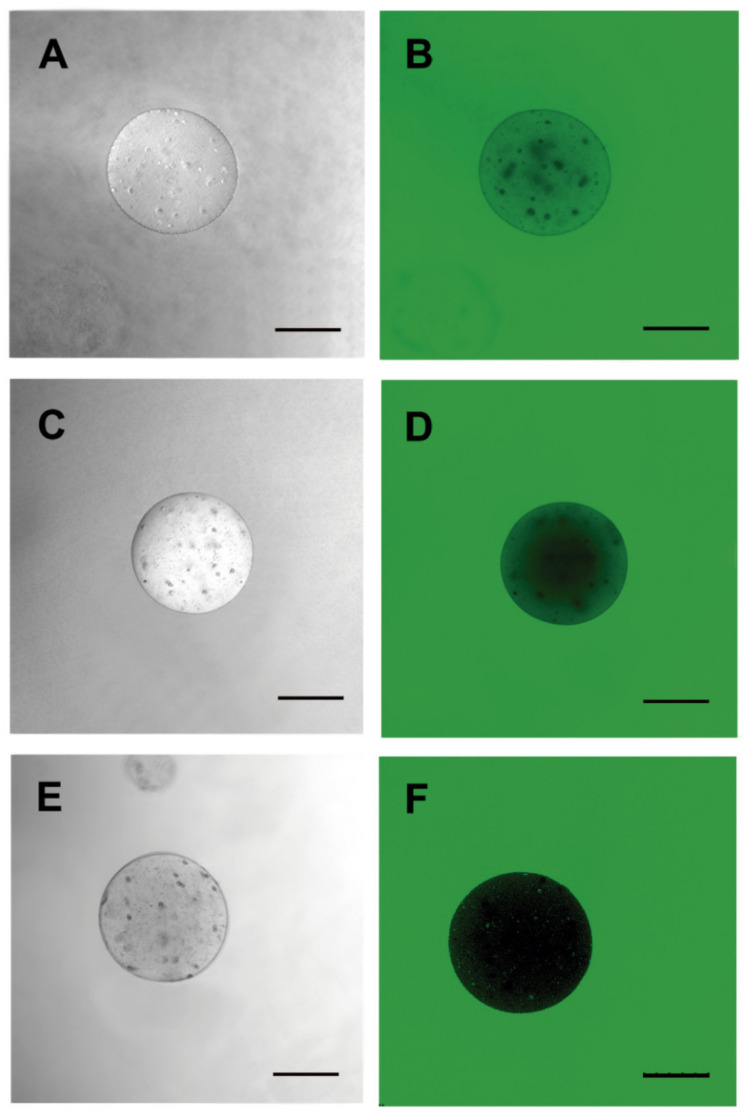
Porosity evaluation of the 3% sodium alginate membrane. Microcapsules containing HepG2 cells using fluorescein isothiocyanate (FITC)-dextran of different molecular weights were incubated for membrane permeability assessments. (**A**,**D**) 20 KDa; (**B**,**E**) 70 KDa and (**C**,**F**) 150 KDa. Images were obtained by fluorescence confocal microscopy and DIC (LEICA). Three independent experiments were carried out in triplicate. Bar 250 µm.

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
