# Peer review of "A Low-Cost Open Source Device for Cell Microencapsulation"

_materials, 2020, doi:10.3390/ma13225090_

Round 1
Reviewer 1 Report
Microencapsulation is a well-studied technique concerning cell therapy and tissue bioengineering techniques to simulate an immune-privileged site, protecting encapsulated cells from the immune system.
In this paper, to achieve 500 µm capsules, two low cost main techniques based on Venturi effect has been developed based on commercially available syringes or 3D printer devices to maintain cells viability and functionality. As a coating was used the low cost sugar based polymer alginate (biocompatible, biodegradable, nontoxicity), composed of 1,4'-β-D-mannuronic acid (M) and α-L-guluronic acid (G) units, that undergoes simple gelation when divalent cations interact with G monomers, forming ionic bridges between adjacent alginate chains. Studies related to viability after 24 h has been performed and porosity studies to demonstrate that this microcapsule displays porosity up to 70 KDa, which is supported by the previous investigations that uses alginate also.
Major revision:
-Although this is a nice device that could allow for the development of new studies concerning microencapsulation for biomedical applications without using toxic immunosuppressive medication, some comments would need to be addressed by the authors:
1-In the device the pressure when pushing the syringe has any impact on the number of capsules/morphology/size/stability/viability?
2-The title should be changed, since no cell transplantation has been accomplished using this device.
3-Further studies concerning longer viability and stability studies should be carried out, since in transplantation procedures a continuous presence of the microcapsules is needed, usually many weeks to avoid transplant rejection.
4-More details about the combination of G alginate and M alginate should be added and discussed to point out the benefits in terms of stability and swelling.
Author Response
Response to Reviewer 1 Comments
The authors would like to thank the reviewers for their time and their valuable comments concerning our manuscript entitled “A Low Cost Open Source Device for Cell Microencapsulation”, which we are resubmitting for consideration of publication. As you can see, we have followed all reviewer suggestions. The critiques were addressed, and changes highlighted in red in the manuscript. I hope this manuscript will now be considered suitable for publication.
Reviewer’s comments:
Reviewer #1
- In the device the pressure when pushing the syringe has any impact on the number of capsules/morphology/size/stability/viability?
Response: In our system, the operator does not push syringe plunger, which is manually controlled through a scroll with small steps. If the operator scrolls fast, this can influence the size and number of capsules. These variables depend on the manipulator, but are easily adjustable, as noted by the few differences among the operators that set this parameter as constant with the scroll plunger. We used 1 mL syringes in our device, but larger diameter syringes (3 mL and 5mL) exhibited significant capsule modifications. The morphology will depend mainly on the concentration and quality of the sodium alginate. Stability and viability are not influenced by operator pressure on the syringe (1 mL) when we obtain round capsules. Our main variable was the gas pressure in the system.
- The title should be changed, since no cell transplantation has been accomplished using this device.
Response: We agree with your excellent suggestion. We have modified the title to: “A Low Cost Open Source Device for Cell Microencapsulation.”
- Further studies concerning longer viability and stability studies should be carried out, since in transplantation procedures a continuous presence of the microcapsules is needed, usually many weeks to avoid transplant rejection.
Response: The main focus of this study is to produce a low cost device to encapsulate cells. To perform in vivo experiments, we intend to use Ultrapure Alginate, applying the protocol published by Paredes-Juarez and co-workers (Materials 7, 2087-2103). However, the results of the viability of microencapsulated cells that grew in the capsules at up to 72 hours of three independent experiments performed in triplicate were added to the supplementary material.
- More details about the combination of G alginate and M alginate should be added and discussed to point out the benefits in terms of stability and swelling.
Response: We have commented and discussed this, as suggested.

Reviewer 2 Report
I strongly believe that the concept of “open source equipment and medical devices” is a very good idea. I think that it will have a big impact, not only on society but also on science. Especially in a reality dominated by pharmaceutical companies focused on profits.
Gas disruption of a naturally flowing column of viscous liquid is a well-known drop formation method. The biggest problem here, however, is the issue of the repeatability of the size of the obtained objects, resulting from the instability of the improvised systems. Even with maximum care in performing , fluctuations are large and we often get mixtures of capsules and imperfect objects. I personally experienced it the hard way when working with carrageenan. Authors propose a solution to this problem by precisely defining the geometry of the device and performing productions on a 3D printer, which ensures not only the repeatability of the geometry but also a low cost of production of one device. The authors pay a lot of attention to show that the repeatability of the capsule size is constant, even taking into account such variables as the different behavior of individual operators. Work is a new refreshing approach to not a new method and application, but the matter of intentionally considering costs adds value to work.
Some bad news now:
As for the disadvantages the selected cell line is problematic for me. HepG2 has very little to do with the original liver cells. They did not retained any of its functions related to detoxification. The authors should comment on the choice of such a line somewhere.
Verse 323 “Electron microscopy is also being performed to better investigate membrane shape and pores.” Where are these photos/ Results?
Verse 351 „To date, two methods have been described for capsule production: 1) based on high voltage dispersion technology….” I would like some literature for annotation 1. Write more (as an alternative technique) on this topic or not writing anything about it.
Verse 356” However, this method is not very exploited, due to its high production cost, while also limiting cell viability and proliferation.” This last information on survival should be supported by some experimental data (can be from literature) that can be compared to what the author shows as his results.
In "DISCUSSION" to verse 363 it is rather information for introduction. "DISCUSSION" should focus on the results the context is important but there is too much of it here.
I have mixed feelings about the work but summing up the advantages that catch my eye on the first reading are positive and it seems to me that the works after corrections should be published.
Author Response
Response to Reviewer 2 Comments
The authors would like to thank the reviewers for their time and their valuable comments concerning our manuscript entitled “A Low Cost Open Source Device for Cell Microencapsulation”, which we are resubmitting for consideration of publication. As you can see, we have followed all reviewer suggestions. The critiques were addressed, and changes highlighted in red in the manuscript. I hope this manuscript will now be considered suitable for publication.
Reviewer #2
- As for the disadvantages the selected cell line is problematic for me. HepG2 has very little to do with the original liver cells. They did not retained any of its functions related to detoxification. The authors should comment on the choice of such a line somewhere.
Response: The main objective of this study was to develop a low cost device for cellular microencapsulation for human cell therapy in liver diseases. In order to minimize animal use, we used the HepG2 strain, a common cellular model for liver functions to standardize our device. However, our main purpose is to microencapsulate freshly isolated cryopreserved primary hepatocytes. We have not performed experiments with primary mouse hepatocytes due to the COVID Pandemic, as only bench work directly associated with COVID is currently being executed in our Institute. According to the literature, a good correlation between cell lines microspheres and primary culture functions is noted.
- Verse 323 “Electron microscopy is also being performed to better investigate membrane shape and pores.” Where are these photos/ Results?
Response: We performed a scanning electron microscopy assessment, but the microsphere pores cannot be visualized (please see the attached file in link https://drive.google.com/file/d/1Y3znC-EeRBHaHFXZ6bLEs-uTgQgmYPLg/view?usp=sharing). We have, thus, removed this comment. After the Pandemic, we intend to perform an adequate characterization of our system
with ultrapure alginate, following the protocol published by Paredes-Juarez and co-workers (Materials 7, 2087-2103).
- Verse 351 „To date, two methods have been described for capsule production: 1) based on high voltage dispersion technology….” I would like some literature for annotation 1. Write more (as an alternative technique) on this topic or not writing anything about it.
Response: We have removed these statements, as recommended.
- Verse 356” However, this method is not very exploited, due to its high production cost, while also limiting cell viability and proliferation.” This last information on survival should be supported by some experimental data (can be from literature) that can be compared to what the author shows as his results.
Response: We are grateful for this observation. We removed this part out for a more fluid discussion.
- In "DISCUSSION" to verse 363 it is rather information for introducti "DISCUSSION" should focus on the results the context is important but there is too much of it here.
Response: The discussion information in sentence 363 was removed and included in the introduction, as suggested.

Round 2
Reviewer 1 Report
All the suggestions were carried out and changes were appropriated